# Specify What?
# A Case-Study using GPT-4 and Formal Methods For Specification Synthesis

**George Granberry** [1]   **Wolfgang Ahrendt** [1]   **Moa Johansson** [1]

## Abstract

Formal specifications are supposed to unambiguously describe the behaviour of (parts of) programs and are usually provided as extra annotations of the program code. The intention is both to document the code and to be able to automatically check compliance of programs using formal methods tools. Writing good specifications can however be both difficult and time-consuming for the programmer. In this case-study, we investigate how GPT-4 can help with the task. We propose a neuro-symbolic integration, by which we augment the LLM prompts with outputs from two formal methods tools in the Frama-C ecosystem (Pathcrawler and EVA), and produce C program annotations in the specifications language ACSL. We demonstrate how this impacts the quality of annotations: information about input/output examples from Pathcrawler produce more context-aware annotations, while the inclusion of EVA reports yields annotations more attuned to runtime errors.

## 1. Introduction

The field of specification synthesis offers a possible solution to the inherent complexities involved in creating and maintaining specifications for software verification. Creating useful specifications demands a deep understanding of both the specification language and the verification process, which can often be as intricate, if not more so, than the software they aim to verify. This complexity poses a significant barrier (Davis et al., 2013; Tyler, 2021), especially in dynamic environments where frequent updates and refactoring are the norm. Maintaining an accurate alignment between ever-evolving code and its specifications can become a cumbersome and error-prone process.

Specification synthesis possibly alleviates these concerns by automating the generation and adaptation of specifications. Instead of requiring developers to manually write detailed specifications – a task that can be both time-consuming and susceptible to human error – specification synthesis aims to infer and edit specifications directly from the codebase and associated context. The goal is to transform specifications into convenient guardrails that provide valuable insights and guidance to programmers, rather than chores performed at the end of the software pipeline.

Early approaches towards generating specifications employed a range of symbolic techniques, encompassing methods like dynamic and static analysis (Lathouwers & Huisman, 2024). For instance, Daikon (Ernst et al., 2007), a widely recognized tool in dynamic analysis, infers properties by observing program behavior at runtime. On the other hand, static analyzers deduce properties based on the program's structure without executing it. Despite their precision, the primary limitation of these methods is their rigidity. Symbolic techniques are constrained by a limited range of expressible properties and typically specialize in specific types of analysis which restricts their flexibility in adapting to diverse verification needs.

On the other side of specification synthesis techniques are the more recent machine-learning-based synthesizers, which include methods like Natural Language Processing (NLP) and Large Language Models (LLMs) (Brown et al., 2020). NLP tools convert documentation and comments into specifications (Blasi et al., 2018) while LLMs stand out for their flexibility and creativity in generating specifications from arbitrary text inputs. These models can theoretically generate any specification that can be articulated in their associated language, provided they are appropriately trained and given the correct prompts.

However, this strength also introduces a significant challenge: the large range of potential specifications LLMs can produce often includes outputs that may not be practically useful. While an LLM can generate a wide array of specifi-

*Equal contribution   [1]Chalmers University of Technology, Gothenburg, Sweden. Correspondence to: George Granberry <georgegr@chalmers.se>, Wolfgang Ahrendt <ahrendt@chalmers.se>, Moa Johansson <moa.johansson@chalmers.se>.

*The first AI for MATH Workshop at the 41st International Conference on Machine Learning, Vienna, Austria. Copyright 2024 by the author(s).*

cations, the lack of inherent direction means that there is no guarantee that the generated specifications will be relevant or valuable for specific verification tasks. This challenge has lead users of LLM-based synthesis to rely on **prompt engineering** (White et al., 2023) techniques in order to increase the likelihood of the LLM to produce specifications that align with their objectives.

In this paper, we introduce a hybrid approach that combines the precision of existing symbolic tools with the flexibility and creative potential of LLMs. By integrating outputs from symbolic analysis of C programs into LLM prompts, this method aims to harness the generative capabilities of LLMs while taking into account the focus and direction of symbolic analysis. As interpreting specifications is subjective, we rely on a human-in-the-loop qualitative analysis to observe patterns in our generated specifications. From a practical sense, we can't always expect our code to be semantically correct when generating a specification for it. Therefore, we also observe how our proposed technique interacts with intentionally incorrect code.

Our findings suggest that this approach does not increase the quantity of generated specifications but rather gives the specifications a focus that mirrors the symbolic analysis of choice. The integration acts as a directive lens, focusing the LLM on generating specifications that align with insights from symbolic analysis, thus yielding specifications that are more relevant to the user who chose said symbolic tools. Each symbolic analysis is interpreted and utilized in a unique way, and some analysis types are given significantly more attention by the LLM than others. In general, these tools are used to both provide some extra context but also increase the likelihood that the LLM will focus on a particular aspect of the specification.

A provided programming including a clear intent plays a crucial role in guiding the LLM during the specification generation process. If the LLM successfully grasps the purpose of the program a programmer *intends* to write, it becomes much more likely to produce specifications that align closely with this intent. However, in the absence of a clearly inferred intent, the LLM tends to default to generating specifications based on surface-level implementation details or the context provided.

## 2. Methods and Tools

**Frama-C and ACSL** The Frama-C ecosystem is an open-source suite of tools designed for the analysis of the source code of software written in C (Kirchner et al., 2015). It integrates various static and dynamic analysis techniques to evaluate the correctness, safety, and security of C programs. It also supports the specification language ACSL (Baudin et al., 2008; Signoles), which is used to formulate *contracts*

consisting of e.g. preconditions – the requirements before a function executes – and postconditions – the expected state after execution. These contracts provide a clear and formal framework for understanding and verifying a function's behavior. Other ACSL annotaions commonly used are *assertions* - stating a condition that needs to be true at some point in execution and *loop invariants* which specify conditions that needs to hold at each iteration of a loop.

**Value Analysis: EVA** The EVA tool uses abstract interpretation to approximate a set of possible values that program variables can take during execution (Blazy et al., 2017). By doing so, it can identify a range of potential issues, such as division by zero, buffer overflows, null pointer dereferences, and arithmetic overflows. EVA's analysis helps in ensuring that the code behaves correctly across all possible execution paths and input values. EVA is designed to respect and work with ACSL annotations when they are present

**Automated Testing: Pathcrawler** The PathCrawler tool is designed for the automated testing of C programs (Williams et al., 2005). Its primary function is to generate and execute test inputs for C code, with a particular focus on achieving high code coverage. Employing a technique known as concolic testing – a combination of concrete and symbolic testing – Pathccrawler efficiently explores different execution paths in the program. This approach not only generates test cases but also executes them, providing valuable information from the execution results across a broad spectrum of program paths. Additionally, PathCrawler allows users to incorporate a test oracle, a mechanism used to classify the outcome of test cases. The test oracle assesses whether the output of a program for a given input is correct or incorrect, aiding in establishing a baseline for "correct" program behavior.

**LLM and Prompts** We have chosen to use GPT-4 (version gpt-4-0125-preview) as our LLM for generating specification. We ran preliminary tests with Gemini as well as GPT-3.5 but found that they returned too many syntactical and semantic errors to draw interesting conclusions from. While open source models such as Llama-3 have recently gained traction, the setup and fine tuning of such a model was out scope for this project, and remain as further work.

We prompt GPT-4 with a C program, instructions for how to generate ACSL annotations (in a step-by-step manner). We also include a few examples of valid annotations in the prompts (see Appendix A). We also experiment with prompts which in addition contain outputs from the EVA and Pathcrawler tools (see Appendix B and C).

## 3. C-program Test Suits

For our study, we have chosen to utilize the 55 programs from the closed-source test suite[1] of Pathcrawler which we will refer to as the **pathcrawler_set**. This suite includes a variety of program types, balancing well-known algorithms like Binary Search with more niche programs such as a Soup Heater controller. It also contains small, specially crafted programs designed to test specific capabilities of Pathcrawler, adding another layer of diversity to our tests. Additionally, **pathcrawler_tests** includes files that provide preconditions to Pathcrawler when it creates test inputs. This was convenient as it saved us from having to provide sensible test inputs for every program that we wanted to use Pathcrawler with. Using a closed-source test suite also has the advantage that at least some of the programs and their annotations are less likley to have appeared in the training data for GPT-4. This test suit helps us test to what extent accurate annotations can be produced for correct programs.

To also investigate if our approach can help with buggy programs, we created a second suite of programs titled **mutated_set**. This comprises 8 of "correct" programs with handcrafted mutations simulating typos, designed to explore a range of programs across two key dimensions: clarity of intent and complexity. To thoroughly study the interactions between these dimensions, this set includes various types of programs: simple programs with clear intent, complex programs with clear intent, simple programs with ambiguous intent, and complex programs with ambiguous intent.

## 4. Generating Annotations

For each program in our two test suits, we generate three sets of ACSL annotations:

1. Using just the program in the prompt (Appendix A)

2. Running EVA on the program and including its report on potential value errors in the prompt shown in Appendix B.

3. Running Pathcrawler on the program, and then including its output about test-cases (input-output pairs) into the prompt shown in Appendix C.

The variability of LLMs like GPT-4 can be adjusted via its temperature setting. As we are interested to to explore a range of potential specifications we choose to generate three distinct specifications for each program (and prompt) within our test suite, repeating the steps above with a temperature setting of 0.7. This approach allows us to capture a spectrum of possible specifications and assess the consistency and variability of the model's output across multiple generations.

[1]Provided to us by the Pathcrawler developers.

## 5. Evaluation

Evaluating specifications is inherently a complex and somewhat ambiguous task, largely due to the absence of a universally correct specification for any given program. Different users often have varying priorities and perspectives on which properties are worth verifying, making the notion of a definitive specification subjective. Similarly, a specification might be logically correct, but more or less trivial with respect to the program at hand, in which case it provides little value.

In light of these challenges, our evaluation methodology does not attempt to benchmark the generated specifications against a predefined gold standard, nor does it aim to determine the optimal approach to creating specifications. Instead, our focus is on identifying the behaviors and patterns that emerge from incorporating symbolic analysis outputs into the specification generation prompts. This approach allows us to better understand the dynamics at play and what kinds of output to expect given a particular prompt.

We propose a primarily qualitative evaluation from two different angles:

- **Types of annotations**: In addition to counting the different types of annotations produced per prompt, we use a human-in-the-loop qualitative analysis to interpret the specification and identify trends depending on which prompt was used, to assess how the different symbolic tool outputs influence the results of the LLM. For this we use the programs in the **pathcrawler_set**.

- **Implementation vs. Intent**: We specifically examine programs in the **mutated_set** to study how errors introduced into the program affect the resultant specifications. This analysis explores how errors, symbolic analyses, intended program functionality, and actual implementation interact.

### 5.1. Types of Annotations

To give an overview, Figure 1 displays the number annotations generated for each annotation type for the three promtps. For all three cases, the most common annotations are unsurprisingly the *requires* and *ensures* statements, which are used to define pre- and post-conditions of functions, followed by *assigns* statements and *loop invariants*.

#### 5.1.1. BASELINE PROMPT

Many of the annotations produced with the baseline prompt were rather simplistic. While not necessarily incorrect or completely useless, these specifications tended to focus on surface-level details of the programs, overlooking deeper, more substantive aspects. This can be seen in Appendix D

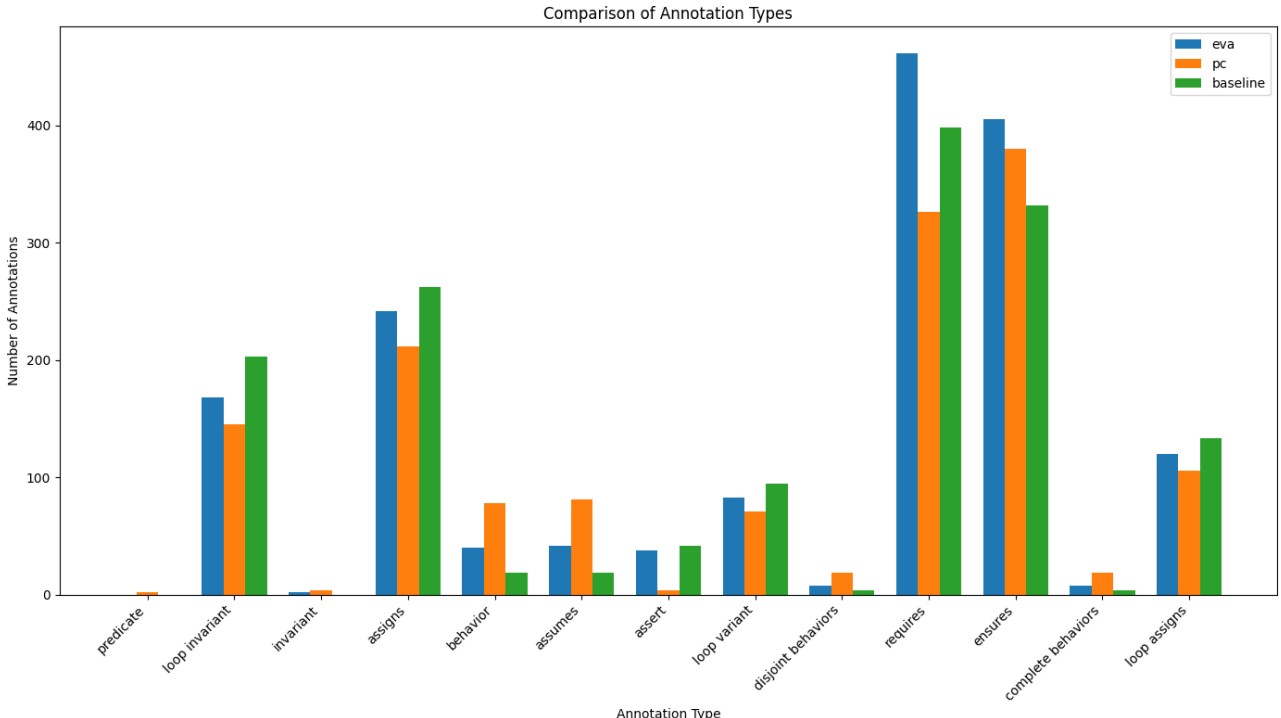

*Figure 1.* Annotation-type counts for each prompt

where the specification completely misses the semantics of binary search.

We note that during an earlier pilot experiment, with a previous version of GPT-4, the same prompt produced a larger number of annotations, both unnecessary but also more substantive, which were missing using a later version. This is exemplified by our preliminary specification generated for binary search shown in Appendix E. This highlights one of the downsides with using GPT-4: users are largely unaware of changes to the product and it is difficult to know what causes changes in behaviour.

### 5.1.2. PROMPT WITH EVA ANNOTATIONS

When given the EVA-augmented prompt, the annotation types generated display distinctive characteristics influenced by the inclusion of EVA reports in the prompt. Notably, these annotations types tend to emphasize aspects that are less about operational semantics and more focused on mitigating runtime errors and defining possible output domains. The manner in which the LLM responds to the information from Eva reports aligns with our expectations. EVA's alarms highlight scenarios where inputs might lead to runtime errors. The LLM addresses these scenarios by generating requires clauses that aim to preclude problematic inputs, effectively setting boundaries to ensure safety. Moreover,

EVA's value analysis provides detailed insights into the range of possible values variables might hold at the end of a function's execution. This data is particularly useful for generating postconditions in the form of ensures clauses. These clauses frequently involve specifying ranges or conditions for variable values at the end of execution, thereby providing useful constraints to the program as a while. However, it seems the LLM has a tendency to often base its specifications primarily on the EVA reports and disregard details of the implementation. As an example, the specification generated for the triangle classifier seen in Appendix F along with its corresponding EVA report, focuses on values from the EVA report while disregarding functional properties of the code.

### 5.1.3. PROMPT WITH PATHCRAWLER INPUT/OUTPUT PAIRS

When using the Pathcrawler-augmented prompt, we observe two main characteristics. First, it produces the highest number of *behavior* clauses. Second, as shown in Appendix G, the quality of the specification is highly dependent on the quantity and quality of the test cases. This leads to specifications with an emphasis on capturing the semantics of program behavior, rather than detailing numerous specific implementation aspects. However, in the case where either quality or quantity of test cases is poor, then the Pathcrawler

output is more of a distraction to the LLM.

This outcome aligns well with our expectations for how Pathcrawler input/output guides the LLM towards considering the broader, more abstract interactions within the program. The input-output pairs showcase how different inputs lead to various outputs, prompting the LLM to produce annotations beyond simple line-by-line code execution and instead focus on the causal relationships and logical flows that define the program's behavior. The weaknesses that we encountered with using Pathcrawler as a tool were a result of Pathcrawler being designed for a different use case rather than any core issue with the methodology of including input/output examples in prompts.

## 5.2. Implementation vs. Intent

What should a specification generation tool do if given a buggy program? Symbolic tools, such as QuickSpec (Smallbone et al., 2017) will simply generate some odd specification to which the buggy program adheres. The user might find this surprising, but it might not be obvious that it is due to a bug. As an LLM can also take cues from function names into account, we speculate that it might be better at inferring the actual *intent* rather than blindly follow the implementation. This in turn might bring to the programmers attention that the program does *not adhere to the specification*, and prompt them to more closely examine the reasons. We therefore test our three prompts on the **mutated_set** of programs.

### 5.2.1. BASELINE PROMPT

We did indeed observe that if the LLM has an intent provided to it, it will create a specification based on that intent even if the buggy implementation contradicts it. On the other hand, annotations in the function body such as loop invariants and asserts tend to still follow the implementation rather than the intent. One strong example of this is the mutated Tritype function shown in figure 2. In this example the implementation and specification are disconnected from each other as the LLM's inferred intent took priority over small yet semantically important implementation details. On the other hand, the code for calculating the Levenshtein distance shown in figure 3 shows us that rather than generating loop annotations from a nuanced understanding of the interactions between implementation and intent, the LLM instead considered the full intent of the function and then added loop annotations only based on nearby implementation. When an obvious bug in the code is detected, the behavior of LLM responds in one of two primary ways. The more common response is that the LLM disregards the error and generate a specification using inferred intent from other language cues in the program. On less frequent occasions the LLM might actively repair the code during the specifi-

```
/*@
  @ requires i >= 0 && j >= 0 && k >= 0;
  @ assigns \nothing;
  @ behavior not_triangle:
  @   assumes i == 0 || j == 0 || k == 0 ||
    i+j <= k || j+k <= i || i+k <= j;
  @   ensures \result == 4;
  @ behavior equilateral_triangle:
  @   assumes i != 0 && j != 0 && k != 0 &&
    i == j && j == k;
  @   ensures \result == 3;
  @ behavior isosceles_triangle:
  @   assumes i != 0 && j != 0 && k != 0 &&
    ((i == j && j != k) || (i != j && j ==
    k) || (i == k && j != k));
  @   ensures \result == 2;
  @ behavior scalene_triangle:
  @   assumes i != 0 && j != 0 && k != 0 &&
    i != j && j != k && i != k && i+j > k
    && j+k > i && i+k > j;
  @   ensures \result == 1;
*/
int tritype(int i, int j, int k){
  int type_code;
  if ((i == 0) || (j == 0) || (k == 0))
    type_code = 4;
  else {
    type_code = 0;
    if (i == j) type_code = type_code + 1;
    if (i == k) type_code = type_code + 2;
    if (j == k) type_code = type_code + 3;
    if (type_code == 0){
      if ((i+j <= k) || (i+k <= i) || (i+k
  <= j))
  type_code = 4;
      else
  type_code = 1;
      }
    else if (type_code > 3) type_code = 3;
    else if ((type_code == 1) && (i+j > k))
     type_code = 2;
    else if ((type_code == 2) && (i+k > j))
     type_code = 2;
    else if ((type_code == 3) && (i+k > i))
     type_code = 2;
    else type_code = 4;
    }
  return type_code;
}
```

*Figure 2.* Mutated Tritype with typos highlighted in red

```
#include <string.h>

/*@ requires \true;
  @ ensures \result <= x && \result <= y &&
    \result <= z;
  @ assigns \nothing;
*/
int min(int x, int y, int z) {
    if (x < y) return (x < z) ? x : z;
    else return (y < z) ? y : z;
}

/*@ requires \valid(s1) && \valid(s2);
  @ ensures \result >= 0;
  @ assigns \nothing;
*/
int levenshtein(char *s1, char *s2) {
    int len1 = strlen(s1), len2 = strlen(s2
    );
    int matrix[len1 + 1][len2 + 1];

    /*@ loop invariant 0 <= x <= len1;
      @ loop assigns matrix[0][0 .. len1];
    */
    for (int x = 0; x <= len1; x++) matrix
    [0][x] = x;

    /*@ loop invariant 0 <= y <= len2;
      @ loop assigns matrix[0 .. len2][0];
    */
    for (int y = 0; y <= len2; y++) matrix[
    y][0] = y;

    /*@ loop invariant 1 <= x <= len1;
      @ loop assigns matrix[1 .. len1][1 ..
     len2];
    */
    for (int x = 1; x <= len1; x++) {
        /*@ loop invariant 1 <= y <= len2;
          @ loop assigns matrix[x][1 ..
    len2];
        */
        for (int y = 1; y <= len2; y++) {
            int cost = (s1[x - 1] == s2[y -
     1]) ? 0 : 1;
            matrix[x][y] = min(matrix[x -
    1][y] + 1, matrix[x][y - 1] + 1, matrix
    [x - 1][y - 1] + cost);
        }
    }

    return matrix[len1][len2];
}
```

*Figure 3.* Mutated Levenshtein distance with typos highlighted in red

cation process, contrary to the instructions to not modify the C code. Quite likely the LLM might have seen very similar programs during training, and the corrected version is simply a more likely continuation.

5.2.2. PROMPTS WITH ADDITIONAL ANNOTATIONS

Adding information from the formal methods tools will not help much in the case of buggy programs. The inclusion of an Eva report continues to steer the LLM's towards specifications focusing on value domains, ignoring any underlying intent of the program, much like observations from bug-free scenarios. Similarly for Pathfinder, input/output pairs did not consistently aid the LLM in recognizing buggy code within a program, even when an oracle was provided specifying which test cases failed. Instead, the specifications largely followed cues from names in the program, as in the baseline case.

## 6. Conclusion

We have conducted a case-study to assess the abilities of GPT-4 to generate specifications of simple C-programs, with or without added information from symbolic formal methods tools. While the evaluation is naturally somewhat subjective and limited to working on simple programs, it shows that adding such information appear improve the quality of the annotations, and can steer the LLM to focus on certain types of annotations of interest to the user. We see this as a first step towards building neuro-symbolic verification systems where formal methods tools and LLMs work in tandem.

## Acknowledgments

This work was supported by funding the Wallenberg AI, Autonomous Systems and Software Program (WASP) funded by the Knut and Alice Wallenberg Foundation.

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

## A. Baseline Prompt

Prompt used for generating ACSL without any additional context.

```
You are a LLM that takes the following inputs and returns a C program annotated
    with ACSL annotations.

Inputs:
1. A C program with no ACSL annotations

GOALS:
1. Describe any abstract properties that could be represented as ACSL annotations
2. Generate ACSL annotations based on your analysis of the program
3. Returning a program with no annotation is not a valid solution
4. Do not edit the C code, only add annotations
5. Make sure to describe your thought process behind the annotations
6. Do not skip any code in the returned solution to make it shorter.
7. If you break any of these rules then my family will disown me.

ANNOTATION EXAMPLES:

Examples 1 (single annotation):
/*@ requires low >= 0 && high <= 9; */

Example 2 (block annotation style):
//Only use this style for function headers. Do not use blocks for multiple
    annoations in the function body
/*@
  @ requires low >= 0 && high <= 9;
  @ requires elem >= 0 && elem <= 9;
*/

Example 3 (loops):
/*@
  @ loop invariant low <= high;
  @ loop variant high - low;
*/
while(low <= high)

Example 4 (loop assigns) (loop assigns must be placed before loop variant):
/*@
  @ loop invariant i >= 0 && i <= 3;
  @ loop assigns fa;
  @ loop variant 3 - i;
*/
while(low <= high)

Example 5 (assigns must be in scope):
//This is VALID because x is a parameter that the function contract can see
{valid_assigns}

// this is NOT VALID because x is in the function body and can not be seen by the
    contract
{invalid_assigns}
```

```
FORMAT INSTRUCTIONS:

First describe your reasoning behind the added annotations

Return the annotated c code wrapped in markdown
```c
...
```

START OF INPUT:

```c
{program}
```
```

## B. EVA Augmented Prompt

The edited instructions for generating specifications with EVA reports.

```
...
1. Describe any abstract properties of the input program that could be
   represented as ACSL annotations
2. Analyze the Eva report and describe how the results could be used in
   generating ACSL annotations
3. Generate ACSL annotations based on your analysis of the program and take
   special account of the properties described when analyzing the Eva report
4. Returning a program with no annotation is not a valid solution
5. Do not edit the C code, only annotations
6. Make sure to describe your thought process behind the annotations
7. Do not skip any code in the returned solution to make it shorter.
8. If you break any of these rules then my family will disown me.
...
START OF INPUT:

Program:
```c
{program}
```

Eva Report:
{eva}
...
```

## C. Pathcrawler Augmented Prompt

The edited instructions for generating specifications with Pathcrawler input/outputs.

```
...
GOALS:
1. Describe any abstract properties of the input program that could be
   represented as ACSL annotations
2. Analyze the pathcrawler CSV and describe any patterns that you see that could
   help you understand the behaviors of the program based on given input/output
   pairs
```

```
3. Describe how these behaviors could be used into creating ACSL annotations
4. Generate ACSL annotations based on your analysis of the program and take
   special account of the properties described when analyzing the Pathcrawler
   CSV file
5. Returning a program with no annotation is not a valid solution
6. Do not edit the C code, only add annotations
7. Make sure to describe your thought process behind the annotations
8. Do not skip any code in the returned solution to make it shorter.
9. If you break any of these rules then my family will disown me.
...
START OF INPUT:

Program:
```c
{program}
```

PathCrawler Output:
{csv}
```

## D. Baseline Binary Search

Specification for Binary Search generated with the baseline prompt.

```
/*@
  @ requires \valid(A + (0..9)) && \valid(&elem);
  @ ensures \result == 0 || \result == 1;
  @ assigns \nothing;
*/
int testme( int A[10], int elem) {
  int low, high, mid, ret ;
  low = 0 ;
  high = 9 ;
  ret = 0 ;
  /*@
    @ loop invariant high >= low;
    @ loop assigns low, high, mid, ret;
    @ loop variant high - low;
  */
  while( ( high > low ) )
    { mid = (low + high) / 2 ;

      if( elem == A[mid] )
         ret = 1;
      if( elem > A[mid] )
        low = mid + 1 ;
      else
        high = mid - 1;
    }
  mid = (low + high) / 2 ;

  if( ( ret != 1)  && ( elem == A[mid]) )
    ret = 1;

  return ret ;
}
```

# E. Preliminary Baseline Binary Search

Specification generated with an earlier verison of GPT-4

```
/*@
  @ requires \valid(A + (0..9));
  @ requires \forall integer i, j; 0 <= i < j < 10 ==> A[i] <= A[j]; // Array is sorted
  @ assigns \nothing;
  @ ensures \result == 1 <==> \exists integer i; 0 <= i < 10 && A[i] == elem;
*/
int testme( int A[10], int elem) {
  int low, high, mid, ret ;

  /*@
    @ assigns low;
  */
  low = 0 ;

  /*@
    @ assigns high;
  */
  //@assert true;
  high = 9 ;

  /*@
    @ assigns ret;
  */
  ret = 0 ;

  /*@
    @ loop invariant 0 <= low <= high <= 9;
    @ loop assigns low, high, mid, ret;
    @ loop variant high - low;
  */
  while( ( high > low ) )
    {
      /*@
        @ assigns mid;
      */
      mid = (low + high) / 2 ;

      /*@
        @ assigns ret;
      */
      if( elem == A[mid] )
          ret = 1;

      /*@
        @ assigns low;
      */
      if( elem > A[mid] )
        low = mid + 1 ;
      else
      /*@
        @ assigns high;
      */
        high = mid - 1;
    }

  /*@
    @ assigns mid;
  */
  mid = (low + high) / 2 ;

  /*@
```

```
    @ assigns ret;
  */
  if( ( ret != 1)  && ( elem == A[mid]) )
    ret = 1;

  /*@
    @ assigns \nothing;
  */
  return ret ;
}
```

## F. LabelsTritype Augmented with an Eva Report

LabelsTritype is a version of Tritype that is meant to test the labeling feature of Pathcrawler. The following is a specification generated for LabelsTritype with an Eva report added.

```
/*@ requires Side1 > 0 && Side2 > 0 && Side3 > 0;
  @ ensures \result >= 1 && \result <= 4;
  @ assigns \nothing;*/
int testme (int Side1, int Side2, int Side3){
   int triOut;
   /*@ assert Side1 <= 2147483647 && Side2 <= 2147483647 && Side3 <= 2147483647; */
   if (Side1 <= 0 || Side2 <= 0 || Side3 <= 0)
   {
      triOut = 4;
      return (triOut);
   }
   triOut = 0;
   if (Side1 == Side2){ triOut = triOut + 1;}
   if (Side1 == Side3){ triOut = triOut + 2;}
   if (Side2 == Side3){triOut = triOut + 3;}
   if (triOut == 0)
   {
     /*@ assert Side1+Side2 <= 2147483647 && Side1+Side3 <= 2147483647 && Side2+Side3 <=
    2147483647; */

     if (Side1+Side2 <= Side3 || Side2+Side3 <= Side1 || Side1+Side3 <= Side2){triOut =
    4;}
     else{triOut = 1;}
     return (triOut);
   }
   if (triOut > 3){triOut = 3;}
   else{
     /*@ assert Side1+Side2 <= 2147483647 && Side1+Side3 <= 2147483647 && Side2+Side3 <=
    2147483647; */
     if (triOut == 1 && Side1+Side2 > Side3){triOut = 2;}
     else{
       if (triOut == 2 && Side1+Side3 > Side2){triOut = 2;}
       else{
         pathcrawler_label(triOut == 3);
         pathcrawler_label(triOut != 3);
         pathcrawler_label(Side2+Side3 > Side1);
         pathcrawler_label(Side2+Side3 <= Side1);
         if (triOut == 3 && Side2+Side3 > Side1){triOut = 2;}
         else{triOut = 4;}
       }
     }
   }
   return (triOut);
}
```

The EVA report provided to the prompt:

```
[kernel] Parsing temp_files/tmpifalg4bf/eva_temp.c (with preprocessing)
```

```
[kernel:typing:implicit-function-declaration] temp_files/tmpifalg4bf/eva_temp.c
   :20: Warning:
  Calling undeclared function pathcrawler_label. Old style K&R code?
[eva] Option -eva-precision 7 detected, automatic configuration of the analysis:
  option -eva-min-loop-unroll set to 0 (default value).
  option -eva-auto-loop-unroll set to 256.
  option -eva-widening-delay set to 4.
  option -eva-partition-history set to 1.
  option -eva-slevel set to 250.
  option -eva-ilevel set to 128.
  option -eva-plevel set to 300.
  option -eva-subdivide-non-linear set to 140.
  option -eva-remove-redundant-alarms set to true (default value).
  option -eva-domains set to 'cvalue,equality,gauges,octagon,symbolic-locations'.
  option -eva-split-return set to 'auto'.
  option -eva-equality-through-calls set to 'formals' (default value).
  option -eva-octagon-through-calls set to true.
[eva] Splitting return states on:
[eva] Analyzing an incomplete application starting at testme
[eva] Computing initial state
[eva] Initial state computed
[eva:initial-state] Values of globals at initialization

[kernel:annot:missing-spec] temp_files/tmpifalg4bf/eva_temp.c:20: Warning:
  Neither code nor specification for function pathcrawler_label, generating
   default assigns from the prototype
[eva] using specification for function pathcrawler_label
[eva:alarm] temp_files/tmpifalg4bf/eva_temp.c:67: Warning:
  signed overflow. assert Side1 + Side2 <= 2147483647;
[eva:alarm] temp_files/tmpifalg4bf/eva_temp.c:69: Warning:
  signed overflow. assert Side1 + Side3 <= 2147483647;
[eva:alarm] temp_files/tmpifalg4bf/eva_temp.c:71: Warning:
  signed overflow. assert Side2 + Side3 <= 2147483647;
[eva:alarm] temp_files/tmpifalg4bf/eva_temp.c:100: Warning:
  signed overflow. assert Side1 + Side2 <= 2147483647;
[eva:alarm] temp_files/tmpifalg4bf/eva_temp.c:112: Warning:
  signed overflow. assert Side1 + Side3 <= 2147483647;
[eva:alarm] temp_files/tmpifalg4bf/eva_temp.c:124: Warning:
  signed overflow. assert Side2 + Side3 <= 2147483647;
[eva] done for function testme
[eva] ====== VALUES COMPUTED ======
[eva:final-states] Values at end of function testme:
  triOut in {1; 2; 3; 4}
  __retres in {1; 2; 3; 4}
[eva:summary] ====== ANALYSIS SUMMARY ======
  ----------------------------
  1 function analyzed (out of 1): 100% coverage.
  In this function, 73 statements reached (out of 73): 100% coverage.
  ----------------------------
  Some errors and warnings have been raised during the analysis:
    by the Eva analyzer:      0 errors    0 warnings
    by the Frama-C kernel:    0 errors    2 warnings
  ----------------------------
  6 alarms generated by the analysis:
```

```
      6 integer overflows
  ------------------------------
  No logical properties have been reached by the analysis.
```

## G. Apache Pathcrawler

Specification generated for the Apache program with Pathcrawler input/output pairs as additional context

```c
#include "apache.h"

/*@
  @ requires \valid(s);
  @ ensures \result >= 0;
*/
unsigned mystrlen(char *s)
{
  int i;
  i = 0;
  /*@
    @ loop invariant \valid(s+i);
    @ loop assigns i;
  */
  while (s[i] != EOS)
    ++i;
  return i;
}

/*@
  @ requires \valid(s1) && \valid(s2) && n >= 0;
  @ ensures \result <= 0 || \result == 0 || \result > 0;
*/
int mystrncmp (const char *s1, const char *s2, int n)
{
  int i;
  int retval;
  i = 0;
  /*@
    @ loop invariant \valid(s1+i) && \valid(s2+i) && i <= n;
    @ loop assigns i, retval;
  */
  do {
    retval = s1[i] - s2[i];
    if (i >= n-1) return retval;
    if (retval != 0) return retval;
    if (s1[i] == EOS) return 0;
    i++;
  } while (1);
}
/*@
  @ requires scheme >= 0 && \valid(uri);
  @ assigns uri[0..TOKEN_SZ];
*/
void testme (char *uri, int scheme)
{
  int cp;
  int c,i;
  char LDAP[5]={"ldap"};
  char *token[TOKEN_SZ];
  if (scheme == 0
      || mystrlen(uri) < scheme) {
    return;
  }
  cp = scheme;
```

```
  if (uri[cp-1] == '/') {

    while (uri[cp] != EOS
           && uri[cp] != '/') {
      ++cp;
    }
    if (uri[cp] == EOS || uri[cp+1] == EOS) return;
    ++cp;
    scheme = cp;
    if (mystrncmp(uri, LDAP, LDAP_SZ) == 0) {
      c = 0;
      token[0] = uri;

      while (uri[cp] != EOS
             && c < TOKEN_SZ) {
        if (uri[cp] == '?') {
          ++c;
          token[c] = uri + cp + 1;
          uri[cp] = EOS;
        }
        ++cp;
      }
      return;
    }
  }
  return;
}
```

The Pathcrawler Input/Output pairs provided. Notice how no output is provided. Since Apache is a void function with side-effects, the input output pairs say nothing about the program.

```
input_scheme,input_uri[0],...,input_uri[14],output,verdict
1,47,47,0,0,0,0,0,0,0,0,0,0,0,0,0,,unknown
1,47,58,0,0,0,0,0,0,0,0,0,0,0,0,0,,unknown
2,108,47,47,47,0,0,0,0,0,0,0,0,0,0,0,,unknown
1,47,0,0,0,0,0,0,0,0,0,0,0,0,0,0,,unknown
5,108,100,97,112,47,47,63,0,0,0,0,0,0,0,0,0,unknown
5,108,100,97,112,47,47,63,47,63,63,0,0,0,0,0,,no_extra_coverage
5,108,100,97,112,47,47,63,63,63,0,0,0,0,0,0,,no_extra_coverage
0,0,0,0,0,0,0,0,0,0,0,0,0,0,0,0,,unknown
422214939,0,0,0,0,0,0,0,0,0,0,0,0,0,0,0,,unknown
1,47,47,47,0,0,0,0,0,0,0,0,0,0,0,0,,unknown
1,58,0,0,0,0,0,0,0,0,0,0,0,0,0,0,,unknown
5,108,100,97,112,47,47,63,63,63,47,0,0,0,0,0,,unknown
5,108,100,97,112,47,47,63,63,47,63,0,0,0,0,0,,no_extra_coverage
4,108,100,97,47,47,47,0,0,0,0,0,0,0,0,0,,unknown
5,108,100,97,112,47,47,47,0,0,0,0,0,0,0,0,,unknown
422214939,47,0,0,0,0,0,0,0,0,0,0,0,0,0,0,,unknown
5,108,100,97,112,47,47,47,63,63,63,0,0,0,0,0,,no_extra_coverage%
```

