# OpenReview forum: "Specify What? A Case-Study using GPT-4 and Formal Methods For Specification Synthesis"
_ICML.cc/2024/Workshop/AI4MATH — ICML 2024 Workshop AI4MATH Poster_

### Official Review · Reviewer_SGrT · 2024-06-11

**Rating:** 5
**Confidence:** 4

**Summary:**

The paper presents a case study investigating the use of GPT-4,  in conjunction with formal methods tools for synthesizing software specifications. The authors propose a neuro-symbolic integration approach that enhances LLM prompts with outputs from two formal methods tools within the Frama-C ecosystem, Pathcrawler and EVA, to produce C program annotations in the ACSL specification language. The study examines how this integration impacts the quality of annotations, suggesting that it leads to more context-aware and runtime error-focused annotations. The paper also explores the approach's effectiveness in dynamic environments with frequent code updates and refactoring.

**Questions:**

- How does the proposed approach scale with the size and complexity of the software project and tested on different LLMs ?
- How does the approach handle ambiguous or incomplete specifications in complex software systems?

**Reasons To Accept:**

- The paper presents a cutting-edge neuro-symbolic integration that skillfully merges the precision of formal methods with the natural language processing capabilities of large language models (LLMs), aiming to reduce the complexity and time investment typically associated with crafting high-quality prompts.
- Case studies are conducted through a controlled experiment utilizing two test suites, establishing a solid foundation for the results. Such findings have the potential to shape the evolution of software engineering tools and methodologies, particularly in the realm of automated specification synthesis.

**Reasons To Reject:**

- The paper's evaluation methodology is qualitative and subjective, casting doubt on the ability to draw definitive conclusions regarding the effectiveness of the integrated approach.
- The investigation is limited to C programming, which might limit its relevance to different programming languages or diverse contexts. Moreover, testing on GPT-4 leaves questions about the universality and robustness of the results across less advanced LLMs.

---

### Official Review · Reviewer_gxB8 · 2024-06-13

**Rating:** 5
**Confidence:** 3

**Summary:**

This paper focus on the specification synthesis task. They propose a approach which integrating the outputs from symbolic analysis of C programs into LLM prompts and generating the specification with LLMs. The motivation is to harness the generative capabilities of LLMs while taking into account the focus and direction of symbolic analysis. In the experiments, a closed-source test suite of Pathcrawler which consists of 55 programs is used as the dataset. 3 different prompting methods are compared: baseline prompt, prompt with EVA annotations, and  prompt with Pathcrawler input/output pairs. The behaviors and patterns of different methods are evaluated from two angles: types of annotations produced, and how errors introduced into the program affect the generated specifications.

**Questions:**

Please refer to the Reasons To Reject part.

**Reasons To Accept:**

- Integrating the outputs from symbolic analysis of C programs into LLM prompts is a new approach.

**Reasons To Reject:**

- The experimental observations does not show significant benefits of incorporating symbolic analysis into the LLM prompts. It is argued that the different focus of symbolic analysis may lead to difference in the generated specifications, however, intuitively, the focus of the LLM can also be effected by choosing different in-context examples.

---

### Meta-Review · Area_Chair_V9DC · 2024-06-13

**Recommendation:** Accept (Poster)
**Confidence:** 4

**Metareview:**

This paper presents a case study of using GPT-4 for the specification synthesis task. Although it is not directly related to math problems, there is a deep connection between formal verification, theorem proving and code generation / program synthesis. It can thus serve as a good complement to this workshop in this direction. The reviewers have pointed out a few flaws of the paper so we believe that the authors will take them into consideration and improve this work for future publication. I recommend to accept.

---

### Decision · Program_Chairs · 2024-06-13

Accept (Poster)